## [Decision Letter · Decision Letter 0]

20 Nov 2024

PONE-D-24-44661Large language models can consistently generate high-quality content for election disinformation operationsPLOS ONE

Dear Dr. Williams,

Thank you for submitting your manuscript to PLOS ONE. After careful consideration, we feel that it has merit but does not fully meet PLOS ONE’s publication criteria as it currently stands. Therefore, we invite you to submit a revised version of the manuscript that addresses the points raised during the review process.

We look forward to receiving your revised manuscript.

Kind regards,

Carlos Carrasco-Farré

Academic Editor

PLOS ONE

**Journal Requirements:**

3. Please note that your Data Availability Statement is currently missing the repository name. If your manuscript is accepted for publication, you will be asked to provide these details on a very short timeline. We therefore suggest that you provide this information now, though we will not hold up the peer review process if you are unable.

**Additional Editor Comments:**

Dear Authors,

Thank you for submitting your manuscript to PLOS ONE. After a thorough evaluation by two expert reviewers, we appreciate the rigor and relevance of your work in examining LLM compliance within election disinformation contexts and the introduction of the DisElect dataset. This study addresses critical issues around AI safety and election integrity, making it a valuable contribution to the field.

Following the reviewers' comments, I recommend a major revision. However, I am confident that with the proposed enhancements, your work will meet the journal’s standards for publication. Below, I summarize the feedback for your attention:

Major Points for Revision

1. Clarification on Model Refusal Patterns and Training Data Bias:

Reviewer 2 suggests a deeper examination of how different outcomes in model refusal may reflect underlying biases in training data. Expanding on this could add further depth to the implications of your findings.

2.Consideration of Prompt Engineering and Red-Teaming:

Additional exploration of prompt engineering’s impact on model compliance and humanness perception is encouraged. This would help illuminate the potential for misuse and how variations in prompting might affect results.

3. Inclusion of a Human Evaluation Element:

Reviewer 2 raised concerns about relying solely on LLMs for "judge" evaluations, as this introduces the possibility of LLM biases influencing conclusions. Adding a human evaluation element could strengthen the study’s validity and might refine the conclusions.

Minor Issues (Textual Clarifications and Formatting):

4. Reviewer 1 noted several minor issues (e.g., typographical errors, inconsistent hyphen use, clarification of terms like "OR," and improved wording on certain passages). Addressing these will enhance the paper’s clarity.

5. Figure 1 ordering: Consider reorganizing the LLMs or clarifying the current ordering criteria.

6. Timeline of model release dates: Including a timeline figure or table could improve the visualization of model “humanness” as it correlates with release dates.

I hope that these suggestions will support the finalization of your study, which I believe has strong potential for publication following these revisions. I appreciate your efforts in addressing these points and look forward to receiving your revised manuscript.

Kind regards,

Carlos Carrasco-Farré

Reviewers' comments:

Reviewer's Responses to Questions

**Comments to the Author**

1. Is the manuscript technically sound, and do the data support the conclusions?

Reviewer #1: Yes

Reviewer #2: Partly

2. Has the statistical analysis been performed appropriately and rigorously? 

Reviewer #1: Yes

Reviewer #2: Yes

3. Have the authors made all data underlying the findings in their manuscript fully available?

Reviewer #1: Yes

Reviewer #2: Yes

4. Is the manuscript presented in an intelligible fashion and written in standard English?

Reviewer #1: Yes

Reviewer #2: Yes

5. Review Comments to the Author

**Reviewer #1:** Overview:

This paper presents a comprehensive two-part investigation into the capabilities of LLMs to generate content for election disinformation operations. The study introduces DisElect, a novel evaluation dataset designed to test LLM compliance with requests to generate election disinformation content. The authors also conduct behavioral experiments to assess how well humans can distinguish between AI-generated and human-written disinformation content. The findings reveal concerning capabilities of current LLMs. In terms of model compliance, most tested LLMs readily comply with requests to generate election disinformation content. A few showed significant refusal rates for malicious prompts. The study uncovered interesting refusal patterns, where models that refuse malicious content tend to also refuse benign election-related prompts and show higher refusal rates for right-wing perspectives compared to left-wing ones. Regarding human detection capabilities, the authors show that almost all models released since 2022 produce content that humans cannot reliably distinguish from human-written content. Some even achieved "above-human-humanness," meaning their generated content was more likely to be perceived as human-written than actual human-written content. The study also demonstrates that LLM-based disinformation operations could be conducted at a fraction of the cost of traditional human-based operations. These findings have significant implications for AI safety, election integrity, and the future of online disinformation. The results suggest that current safety measures in LLMs may be insufficient to prevent their potential misuse in election disinformation campaigns.

Strengths:

• Rigorous methodology combining systematic model evaluation with human experiments

• Comprehensive testing across multiple LLMs spanning different sizes and release dates

• Novel focus on hyperlocal election contexts

• Thorough statistical analysis of results

• Important implications for AI safety and election integrity

Major Issues:

Having carefully reviewed the manuscript, I do not find significant issues that would require major revisions. The paper is methodologically sound and comprehensive within its stated scope. The authors have demonstrated appropriate awareness of the study's limitations, such as the limited prompt engineering exploration and English-language focus.

Minor Issues:

• Page 5: “Weidinger et al. [25] propose a sociotechnical approach these safety evaluations, consisting of evaluations at three intersecting levels”

• Page 8: Inconsistent use of hyphens in the sentence discussing UK context

• Page 15: The sentence about GPT-3.5 Turbo and Mistral's refusal patterns needs clarification. Consider rewording to better explain the relationship between refusals in DisElect.BL versus DisElect.VT/MP

• Fig.1: The ordering principle for LLMs in the visualization is not immediately clear. Consider explicitly stating the ordering criteria or reorganizing based on a clear principle (e.g., chronological order or by company)

• The paper mentions the release date of LLMs multiple times and finds it correlated with the humanness. However, there is no specific presentation of this. Consider adding a timeline figure or table showing this.

• Page 19: The discussion of GPT-3.5 Turbo's consistent performance across stages (std = .03) would be more compelling with a supplementary figure showing mean and standard deviation of performance across stages for all LLMs

• Page 23: Typographical error "LLms" should be "LLMs"

• Page 24: Abbreviation "OR" is used before being defined as "Odds Ratio"

**Reviewer #2: **The manuscript presents a technically sound evaluation of LLM compliance using the DisElect dataset. The methodology for constructing the dataset, including prompt templates and variable selection, is well-defined.

Strengths

- The study addresses a timely and important topic with potential societal implications.

- The DisElect dataset is a valuable contribution to the field of AI safety evaluation.

- The authors provide a comprehensive discussion of their findings and acknowledge the limitations of their study.

Areas for Improvement

- While the authors acknowledge the limited number of LLMs evaluated, a more thorough discussion on what the difefrent outcomes of model actually mean would be a goo addition. Since that might point towards training data biasness.

- Further investigation into the impact of prompt engineering and red-teaming on LLM compliance and humanness perception would provide a more complete understanding of the potential for malicious use. This I beleive will be very critical in future usage of these kind of methods

- I have some reservations of using LLM as judge method. Specifically for these kind of study, it would also mean we are trusting another LLM to validate a "perception" of biasness form another LLMs output. A human evaluation element i believe not only strengthens the paper but might be necessary. I'll be very happy to see such an element added and if that affects the papers conclusions.

If this is done, then I have no reservations recommending the manuscript for publication in PLOS ONE after the authors have addressed the areas for improvement.

6. PLOS authors have the option to publish the peer review history of their article (what does this mean?). If published, this will include your full peer review and any attached files.

Reviewer #1: No

Reviewer #2: **Yes: **Rabimba Karanjai

---

## [Author Response · Author response to Decision Letter 1]

5 Dec 2024

Editor:

Major Points for Revision

1. Clarification on Model Refusal Patterns and Training Data Bias: See response to reviewer 2 comment 1

2. Consideration of Prompt Engineering and Red-Teaming: See response to reviewer 2 comment 2

3. Inclusion of a Human Evaluation Element: See response to reviewer 2 comment 3

Minor Issues:

4. Typographical errors, inconsistent hyphen use, clarification of terms like "OR," and improved wording on certain passages: See response to reviewer 1 comments 1,2,3,7,8

5. Figure 1 ordering: See response to reviewer 1 comment 4

6. Timeline of model release dates: See response to reviewer 1 comment 5

___

Reviewer 1:

Comment 1: Typo highlighted on page 5: “Weidinger et al. [25] propose a sociotechnical approach these safety evaluations, consisting of evaluations at three intersecting levels”

Response: Thank you, we have added “to” in between “approach” and “these”.

Comment 2: Page 8: Inconsistent use of hyphens in the sentence discussing UK context

Response: Thank you, we have addressed this.

Comment 3: Page 15: The sentence about GPT-3.5 Turbo and Mistral's refusal patterns needs clarification. Consider rewording to better explain the relationship between refusals in DisElect.BL versus DisElect.VT/MP

Response: We appreciate this discussion may not have been sufficient, and have expanded on this paragraph to discuss what refusals on Diselect.BL represent.

Comment 4: Fig.1: The ordering principle for LLMs in the visualization is not immediately clear. Consider explicitly stating the ordering criteria or reorganizing based on a clear principle (e.g., chronological order or by company)

Response: Thanks for your concern, we appreciate that the figures and their captions are not displayed together in the materials provided to reviewers so it was a bit hard to grasp this! To help address this, we have included a comment at the beginning of the results section to make clear that models are sorted by release date in all visualisations, as well as emphasising this in the caption.

Comment 5: The paper mentions the release date of LLMs multiple times and finds it correlated with the humanness. However, there is no specific presentation of this. Consider adding a timeline figure or table showing this.

Response: That’s a good idea – we have added Fig. 8 to visualise this.

Comment 6: Page 19: The discussion of GPT-3.5 Turbo's consistent performance across stages (std = .03) would be more compelling with a supplementary figure showing mean and standard deviation of performance across stages for all LLMs

Response: Thank you, this makes sense – We have added Fig. 7 to explicitly visualise standard deviation (both across stages and experiments), and have expanded the discussion on page 19 to accompany this.

Comment 7: Page 23: Typographical error "LLms" should be "LLMs"

Response: Thank you, we have fixed this.

Comment 8: Page 24: Abbreviation "OR" is used before being defined as "Odds Ratio"

Response: Thank you for spotting this, we have addressed this.

___

Reviewer 2:

Comment 1:While the authors acknowledge the limited number of LLMs evaluated, a more thorough discussion on what the difefrent outcomes of model actually mean would be a goo addition. Since that might point towards training data biasness.

Response: Thank you, we agree that understanding the biases of models encoded through training data is an important aspect of the explainability of the results of these models. We have expanded the results section for the DisElect eval, further developing the discussion on our analysis of how refusal rates vary across different parameters such as MP gender and party to link these findings to potential representation imbalances in training data to address this (see page 18).

Comment 2: Further investigation into the impact of prompt engineering and red-teaming on LLM compliance and humanness perception would provide a more complete understanding of the potential for malicious use. This I beleive will be very critical in future usage of these kind of methods

Response: It’s a great point that adversarial prompting is inevitable in malicious use - we think that that our study shows that on this kind of task malicious actors can generate high-impact content without need for any adversarial strategies (because most models don’t refuse, and achieve high-humanness as is), and the work/cost involved for a malicious actor in deploying them is lower without adversarial strategies. We have added a section on adversarial prompting to our literature review (see page 6), and have made more explicit references to and discussions of adversarial strategies throughout this paper to try to address this (page 28, 29, 30).

As you suggest, the potential for using prompt engineering to generate responses that score more highly in humanness is great idea that we think would be a really interesting aspect of future work exploring how humans perceive AI generated content, and have made reference to this in the future work section (page 30).

Comment 3:I have some reservations of using LLM as judge method. Specifically for these kind of study, it would also mean we are trusting another LLM to validate a "perception" of biasness form another LLMs output. A human evaluation element i believe not only strengthens the paper but might be necessary. I'll be very happy to see such an element added and if that affects the papers conclusions.

Response: Propagating bias by using models to evaluate other models is definitely a fundamental concern that we share, in the risks this poses to scientific validity and false perceptions of the safety and fairness of AI models, and we appreciate you raising it.

We do include a human evaluation of our LLM judge approach, presenting performance metrics on a human-annotated sample of model responses. We have expanded our discussion of LLM-as-a-judge as a method for evaluating other LLMs, as it was perhaps not sufficiently well signposted in the first draft (see page 12/13). Our analysis shows that identifying clear patterns or semantics in text, such as whether a model response is refusing to comply with an instruction, seems to be a task that is well suited to an LLM, which is why we deploy it for this part of our study.

We completely agree that more complex, subjective judgements, where bias may play a more significant impact, are much harder to justify using an LLM to evaluate. As such, the latter half of the paper focuses exclusively on human evaluation of LLM outputs to explore how humans perceive this content compared to human-written content, precisely because we believe that evaluating the safety of AI models requires nuanced human perspectives and judgements - we also study the biases of the human participants themselves in this element.

---

## [Decision Letter · Decision Letter 1]

30 Dec 2024

Large language models can consistently generate high-quality content for election disinformation operations

PONE-D-24-44661R1

Dear Dr. Williams,

We’re pleased to inform you that your manuscript has been judged scientifically suitable for publication and will be formally accepted for publication once it meets all outstanding technical requirements.

Kind regards,

Jinran Wu, PhD

Academic Editor

PLOS ONE

Additional Editor Comments (optional):

Reviewers' comments:

Reviewer's Responses to Questions

**Comments to the Author**

1. If the authors have adequately addressed your comments raised in a previous round of review and you feel that this manuscript is now acceptable for publication, you may indicate that here to bypass the “Comments to the Author” section, enter your conflict of interest statement in the “Confidential to Editor” section, and submit your "Accept" recommendation.

Reviewer #1: All comments have been addressed

2. Is the manuscript technically sound, and do the data support the conclusions?

Reviewer #1: (No Response)

3. Has the statistical analysis been performed appropriately and rigorously? 

Reviewer #1: (No Response)

4. Have the authors made all data underlying the findings in their manuscript fully available?

Reviewer #1: (No Response)

5. Is the manuscript presented in an intelligible fashion and written in standard English?

Reviewer #1: (No Response)

6. Review Comments to the Author

Reviewer #1: (No Response)

7. PLOS authors have the option to publish the peer review history of their article (what does this mean?). If published, this will include your full peer review and any attached files.

Reviewer #1: No

---

## [Editor Report · Acceptance letter]

PONE-D-24-44661R1

PLOS ONE

Dear Dr. Williams,

I'm pleased to inform you that your manuscript has been deemed suitable for publication in PLOS ONE. Congratulations! Your manuscript is now being handed over to our production team.

Kind regards,

on behalf of

Dr. Jinran Wu

Academic Editor

PLOS ONE